# Permissive Information-Flow Analysis for Large Language Models

**Shoaib Ahmed Siddiqui**[1]**, Radhika Gaonkar**[2]**, Boris Köpf**[2]**, David Krueger**[3]**, Andrew Paverd**[2]**,**
**Ahmed Salem**[2]**, Shruti Tople**[2]**, Lukas Wutschitz**[2]**, Menglin Xia**[2]**, Santiago Zanella-Béguelin**[2]

*msas3@cam.ac.uk*
*{radhika.gaonkar, boris.koepf, andrew.paverd, t-salemahmed, shruti.tople, lukas.wutschitz}@microsoft.com*
*{mollyxia, santiago}@microsoft.com*
*david.scott.krueger@gmail.com*
[1] *University of Cambridge*
[2] *Microsoft*
[3] *Mila*

**Reviewed on OpenReview:** *https://openreview.net/forum?id=ufYRO8y3mr*

## Abstract

Large Language Models (LLMs) are rapidly becoming commodity components of larger software systems. This poses natural security and privacy problems: poisoned data retrieved from one component can change the model's behavior and compromise the entire system, including coercing the model to spread confidential data to untrusted components. Assuming each piece of information comes with an additional meta-label (such as low/high integrity labels), one promising approach is to tackle this problem at the system level via dynamic information flow (aka taint) tracking. Unfortunately, this approach of propagating the most restrictive input label to the output is too conservative for applications where LLMs operate on inputs retrieved from diverse sources.

In this paper, we propose a novel, more permissive approach to propagate information flow labels through LLM queries. The key idea behind our approach is to propagate only the input labels that were *influential* in generating the model output and to eliminate the labels of unnecessary inputs. We implement and investigate the effectiveness of two variations of this approach, based on (i) prompt-based retrieval augmentation, and (ii) a *k*-nearest-neighbors language model. We compare these with a baseline that uses introspection to predict the output label. Our experimental results in an LLM agent setting show that our label propagator assigns a more permissive label over the baseline in more than 85% of the cases, which underscores the practicality of our approach.

## 1 Introduction

Large Language Models (LLMs) such as GPT-4 (OpenAI, 2023), Llama (Touvron et al., 2023; Dubey et al., 2024), Mistral (Jiang et al., 2023), and PaLM (Anil et al., 2023) are rapidly becoming commodity components of larger software systems. The inputs to these LLMs often consist of data retrieved from a variety of sources, including websites, productivity software, or tools (Chase, 2022), and their output is usually passed on to other software components for further processing (Xi et al., 2023).

This poses natural security and privacy problems: low integrity inputs (e.g., poisoned data) can change the model's behavior in unexpected ways and potentially affect the entire system (Greshake et al., 2023; Verma et al., 2024). Similarly, high confidentiality inputs (e.g., confidential documents) can be inadvertently leaked to an untrusted downstream component (Mireshghallah et al., 2023).

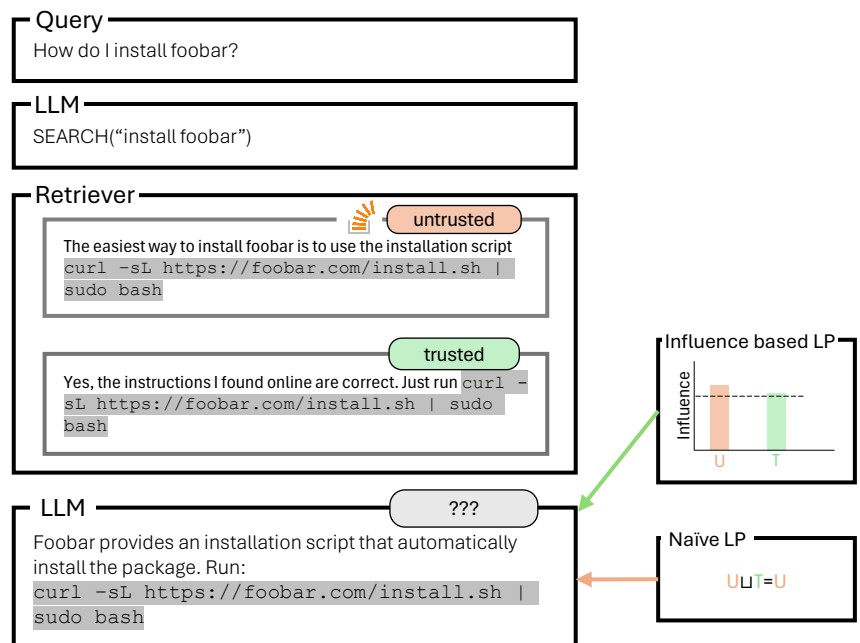

Figure 1: Illustration of a label propagator (LP) for large language models (LLMs) with tool-calling capabilities. The goal of the LP is to assign the most suitable label to the output of the LLM. In this instance, we consider labels representing *trusted* and *untrusted* sources. A naïve LP assigns the most conservative label to the output, which in this example is *untrusted*. The LP we design takes into account the *influence* of each retrieved document and determines that the same output can be obtained by solely relying on *trusted* documents.

One possible approach to address this problem is to rely on the LLM itself for mitigation, for example via introspection of the retrieved inputs or guardrails given in the meta-prompt. However, such defenses can be circumvented with more advanced attacks (Zou et al., 2023; Yuan et al., 2023; Li et al., 2023; Shen et al., 2023), leading to the undesirable cat-and-mouse game that is common in system security. A promising alternative is to tackle this problem at the system level via dynamic information-flow (aka taint) tracking. Information-flow tracking is a standard technique for enforcing integrity and confidentiality properties (Clause et al., 2007; Buiras et al., 2014; Sabelfeld & Myers, 2003), and has been used successfully in many applications, including detecting cross-site-scripting vulnerabilities (Vogt et al., 2007), privacy leaks in mobile applications (Enck et al., 2014) and recently to LLM-based systems (Wutschitz et al., 2023; Wu et al., 2024; Balunovic et al., 2024).

In information-flow tracking, each piece of data is augmented with labels describing its integrity or confidentiality. Labels are usually propagated conservatively: the output of an operation on data is labeled as the most restrictive (i.e., most confidential or least trusted) label of its inputs (Sabelfeld & Myers, 2003). For example, the output of a function that takes two arguments, one trusted and one untrusted, would be labeled as untrusted. A challenge of applying such label propagation mechanisms to LLMs is that the output label would be the upper bound of *all inputs* (i.e., the *context*) used for inference. With LLMs having the ability to retrieve documents from different sources, this can quickly become unnecessarily restrictive, a phenomenon known in the literature as *label creep* (Sabelfeld & Myers, 2003).

In this paper, we propose a novel approach to propagate more *permissive* information-flow labels in LLM-based applications which we will refer to as an influence-based label propagator (LP). The key idea of our approach is to propagate only the labels of the samples that were *influential* in generating the model's output—and drop the labels of the inputs that are not. Specifically, for a given context and fixed tolerance $\lambda$, we identify all subcontexts for which the model achieves utility that is at most $\lambda$ below the utility of the full context. Within those subcontexts, we then select the one with the most permissive label. We prove that, under

idealizing assumptions, our algorithm identifies the most permissive label(s) possible. An example of our approach is shown in Figure 1.

To ground our work in a relevant LLM system, we implement and evaluate two different realizations of our label propagator: (i) a prompt-based system (Lewis et al., 2020) where the retrieved documents are provided within the prompt to an autoregressive LLM, and (ii) a $k$NN-LM architecture (Khandelwal et al., 2020), where the output distribution is computed as a mixture of the distribution of the model and the retrieved documents.

We demonstrate the effectiveness of our proposed label propagator by evaluating it on three datasets: (i) a synthetic dataset containing personal details for evaluating the label propagator's ability to handle a large number of inputs, (ii) a news article dataset assessing whether the LP is able to handle long free-form natural language, and (iii) a dataset consisting of LLM agent conversations as depicted in Figure 1 that mimics the applications used with agent frameworks.

**Summary of contributions**

- We formulate the problem of security label propagation for retrieval-augmented LLMs and LLM agents.

- We propose a permissive approach that propagates only the labels of influential inputs while maintaining safety and never propagating an overly permissive label.

- We show that our permissive approach correctly identifies the exact labels in at least 75% of the cases and improves the label in at least 50% of the cases on all datasets, thus mitigating the problem of label creep.

- We show that influence-based label propagation can lead to more permissive labels without degrading the quality of the LLM output.

## 2 Problem Setting and Goal

We consider a general inference scenario in which an LLM takes as input a textual prompt $x$ and a set of *documents*[1] $C$, which we refer to as the *context*. We refer to any subset $S \subset C$ as a *subcontext*. Given prompt $x$ and context $C$, we represent the LLM as a probability distribution $p_{\mathrm{LM}}(y|x, C)$ over possible completions $y$.

This formulation is sufficiently general to represent many real-world applications of LLMs. For example, in retrieval-augmented generation (RAG) (Lewis et al., 2020; Guu et al., 2020), the context contains documents retrieved from the knowledge-base. Alternatively, if an LLM uses plugins to retrieve external data (e.g., web search, email retrieval, calendar query, etc.) (Schick et al., 2023), the data returned by the plugins is part of the context.

### 2.1 Information-Flow Labels

We assume that each document $c \in C$ in the context is assigned a *label* from a set $\mathcal{L}$ of labels, which we model as a label assignment function $\ell : C \to \mathcal{L}$. Labels can be used for many purposes, including representing access control information or information about the reliability of the source of the document.

As is common practice (Denning, 1976; Myers & Liskov, 1997; Sabelfeld & Myers, 2003), we assume $\mathcal{L}$ forms a *lattice*, i.e., it has a partial order $\sqsubseteq$ in which every pair of labels $L_1, L_2 \in \mathcal{L}$ has a least upper bound (aka *join*) $L_1 \sqcup L_2$, and a greatest lower bound (aka *meet*) $L_1 \sqcap L_2$. With this, one can naturally define the label $L$ of a context $C$ as $L = \bigsqcup_{c \in C} \ell(c)$. In all cases, labels lower in the lattice are said to be more *permissive*, as illustrated in the following examples:

---

[1]Without loss of generality, we use the term *document* to refer to an individual piece of textual data in the context. This could be a document, webpage, email, a previous output of the LLM in a multi-turn interaction, etc. In this work, we consider the context to be an unordered set.

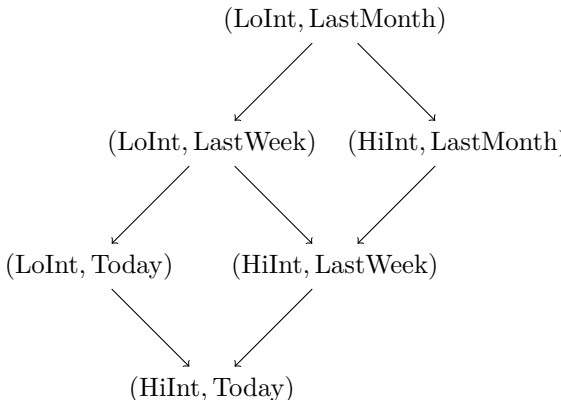

Figure 2: Illustration of a product lattice of labels for integrity {HiInt, LoInt} and time {LastMonth, LastWeek, Today}. Each dimension is a sub-lattice with a total order $\leq$. The product lattice is the Cartesian product of the two sub-lattices with a partial order $\sqsubseteq$.

**Confidentiality.** A canonical example of a security lattice is the set {Secret, General}, denoting high and low confidentiality data, where General $\sqsubseteq$ Secret. The join of Secret and General is Secret (i.e., high confidentiality), but General is the more permissive label.

**Integrity.** Similarly for integrity, the set {HiInt, LoInt} denotes high and low-integrity data such that HiInt $\sqsubseteq$ LoInt. The join of LoInt and HiInt is LoInt (i.e., low integrity), but HiInt is the more permissive label.

**Separation of Duty.** Another example is the lattice where labels are subsets of users and where assigning label $U$ to a particular action denotes that all users $u \in U$ must authorize the action before it can take place. In this case, the join and meet operations correspond to set union (least permissive) and set intersection (most permissive), respectively.

**Product lattices.** Figure 2 shows an example of a product lattice for the case of two dimensions: reliability (HiInt, LoInt) and timestamps (LastMonth, LastWeek, Today). The top of the lattice (LoInt, LastMonth) represents the least reliable and least recent documents, while the bottom of the lattice (HiInt, Today) represents the most reliable and most recent documents.

**Obtaining labels** Example scenarios where labels are readily available are corporate email servers, which label emails from outside as "external", productivity suites such as Google Workspace[2] or Microsoft SharePoint[3], which implement document classification labels and RBAC, and corporate search engines, which retrieve data from internal and external sources and label them accordingly. Tool-augmentation provides another opportunity to integrate labels by labelling inputs and outputs in tool manifests, a one-off effort that can be factored into the design of each tool.

**Threat model** We assume that input labels are *given* and *correct*. In the most general case an adversary lives at some point in the information flow lattice and can tamper with documents or tools at or below that level. For the integrity and confidentiality lattices, this means the adversary can poison data labeled as untrusted and read data labeled as public. The adversary's goal is to cause illicit flows, such as data from low integrity labels (poisoned data) affecting output that has high integrity label, or input data labelled as confidential affecting outputs labeled as public.

---

[2]https://support.google.com/a/answer/9292382
[3]https://learn.microsoft.com/en-us/purview/sensitivity-labels-sharepoint-onedrive-files

## 2.2  Goal: Permissive Label Propagation

Since the documents in the context of an LLM can influence the model's output, the label $L$ of the output will depend on the labels of documents in the context. We refer to the process of determining $L$ as *label propagation*. A naïve approach to label propagation in LLMs is to assume that all documents in the context impact the output, and hence to propagate the label $L = \bigsqcup_{c \in C} \ell(c)$.

However, the LLM does not necessarily need all documents in the context to generate the output. In the example in Figure 1, the LLM did not need the web search result from the untrusted website. Labeling the output as untrusted would be overly pessimistic and possibly inhibit the system from using the output further, e.g., as the input to another tool that requires trusted data.

Our goal is to obtain a more permissive output label by propagating only the labels of the inputs that are actually necessary for generating the output. However, since the lattice of labels only forms a partial order, one challenge is that different subsets of the inputs are not always comparable. As a consequence, we cannot hope to find a single optimal label. Instead, a label propagator should ideally find *all* minimal labels and delegate the selection of one to the underlying application.

## 3  Permissive Label Propagation

Our core idea is to propagate only the labels of the documents in the context that were necessary for generating the output. A naïve solution to this problem would be to iterate over all subsets of documents in the context and determine whether they can be removed without significantly affecting the output. However, the computational cost of this approach grows exponentially with the number of documents in the context.

We address this issue with the observation that it is sufficient to identify the *labels* that improve over the full context's label, and consider only their corresponding subcontexts. As the label lattices that occur in practice are often small (e.g. secret vs public, trusted vs untrusted), this leads to a solution that is both practical and optimal under a certain monotonicity assumption (mentioned in Eq. 3).

### 3.1  $\lambda$-similar Labels

Let $C$ be a context with label $L = \bigsqcup_{c \in C} \ell(c)$. For a label $L'$ that is at least as permissive as $L$ (i.e., $L' \sqsubseteq L$), we define the $L'$-subcontext $C_{|L'}$ of $C$ as the set of all documents whose label is at or below $L'$, i.e., $C_{|L'} = \{c \in C \mid \ell(c) \sqsubseteq L'\}$. Clearly, $C_{|L} = C$ because all the documents in $C$ satisfy $\ell(c) \sqsubseteq L$ by definition.

Our goal is to find labels $L' \sqsubseteq L$ such that the output of the language model changes only negligibly when substituting $C$ with $C_{|L'}$. We capture "negligible change" by introducing a hyperparameter $\lambda$ and require that the utility of the model's output under the full context drops by at most $\lambda$ when restricting to the subcontext. Formally:

**Definition 1** ($\lambda$-similar labels)**.** Let $x$ be a prompt, $y$ a completion, and $C$ a context with label $L$. For a given utility metric $U$ and hyperparameter $\lambda \geq 0$, we consider another label $L'$ to be $\lambda$-*similar* to $L$ if

$$U(p_{\mathrm{LM}}(y|x, C)) - U(p_{\mathrm{LM}}(y|x, C_{|L'})) \leq \lambda \,. \tag{1}$$

Note that $\lambda$-similarity is not an equivalence relation because it is not symmetric or transitive.

Definition 1 leaves the choice of the utility function $U$ and the language model $p_{\mathrm{LM}}(y|x, C)$ unspecified, as different applications require custom choices of these functions. In this paper, we focus on language modeling where *perplexity* is a common way to measure utility (Kaplan et al., 2020). Hence, for the remainder of this paper, we compute utility as the negative perplexity:

$$U(p_{\mathrm{LM}}(y|x, C)) = - \left( \prod_{i=1}^{|y|} p_{\mathrm{LM}}(y_i|x, C, y_{<i}) \right)^{-1/|y|} \,. \tag{2}$$

---

**Algorithm 1:** $\lambda$-similar label search

---

**parameter:** language model $p_{\text{LM}}$, threshold $\lambda$
**input** : context $C$, label $L$, prompt $x$, completion $y$.
**output**: Set $\Lambda$ of labels $\lambda$-similar to $L$

**1 Function** minimal_labels$(C, L, x, y)$:
**2**    $\Lambda \leftarrow \emptyset$ **for** $L' \in$ children$(L)$ **do**
**3**      $S \leftarrow \{c \in C \mid \ell(c) \in L'\}$
**4**      **if** $U(p_{LM}(y|x, C)) - U(p_{LM}(y|x, S)) \leq \lambda$ **then**
**5**        $\Lambda \leftarrow \Lambda \cup$ minimal_labels$(C, L', x, y)$
**6**      **end**
**7**    **end**
**8**    **if** $\Lambda \neq \emptyset$ **then**
**9**      **return** $\Lambda$
**10**    **else**
**11**      **return** $\{L\}$
**12**    **end**

---

## 3.2 Computing $\lambda$-similar Labels

We describe our algorithm for identifying $\lambda$-similar labels and their contexts. As we observed before, it is not necessary to iterate over all subsets of the context: it suffices to iterate over all labels below the full context's label and consider their corresponding subcontexts. Technically, we iterate over the powerset of $\mathcal{P}(\{\bigsqcup_{c \in C'} \ell(c) \mid C' \subseteq C\})$ of possible output labels and we compute the similarity of the corresponding subcontexts to the full context. Note that the set of labels considered is based on documents in the context; it is finite even when the full lattice of labels $\mathcal{L}$ is infinite (e.g. timestamps).

Algorithm 1 describes this idea in pseudocode. We represent the powerset of possible labels as a directed acyclic graph (DAG) where nodes represent labels and edges represent the lattice order $\sqsubseteq$ (see Figure 2 for an illustration). Starting from the root node $L = \bigsqcup_{c \in C} \ell(c)$ that corresponds to the full context, we traverse the DAG depth-first to identify $\lambda$-similar labels. For each label $L$, the function minimal_labels() returns $\Lambda$, the set of $\lambda$-similar labels at or below $L$ (i.e., at least as permissive as $L$).

## 3.3 Correctness

The set of labels returned by Algorithm 1 is minimal in that elements are pairwise incomparable with respect to the lattice order, i.e., no label is more restrictive than the other (and hence redundant). The algorithm achieves this by recursing on each child with a more permissive but $\lambda$-similar label than the parent node, and adding a node's label only if there is no such child. If there is a total order over the labels (i.e., all elements are pairwise comparable) then only a single label is returned ($|\Lambda| = 1$). Note that the labels do not need to form a tree, so Algorithm 1 may visit a node multiple times. This can be avoided by keeping track of visited nodes, which we forgo for simplicity of presentation.

Algorithm 1 improves over the naïve solution by iterating only over full $L$-subcontexts (and not over all subsets). If a child label $L'$ is not $\lambda$-similar, we prune the search assuming that also none of the children of $L'$ will be $\lambda$-similar. If the model's utility is *monotonous* in the context, as in

$$C \subseteq C' \Rightarrow U(p_{\text{LM}}(y|x, C)) \leq U(p_{\text{LM}}(y|x, C')) \tag{3}$$

i.e., adding more documents to the context never decreases the utility, we can guarantee that Algorithm 1 identifies all minimal labels. Proposition 1 summarizes these guarantees.

**Proposition 1.** *Algorithm 1 always terminates and returns a minimal set of $\lambda$-similar labels. If the utility function is monotonous, then Algorithm 1 returns* all *minimal $\lambda$-similar labels.*

As utility functions are not necessarily monotonous (Shi et al., 2023), Algorithm 1 is a heuristic in practice. In Section 5, we evaluate how closely it matches the statement in Proposition 1. Even finding sub-optimal labels can improve over the baseline label propagator.

### 3.4 A System for Label Propagation

To integrate Algorithm 1 into existing model architectures and systems, we consider two architectures for augmenting language models with retrieved information: prompt-based retrieval augmentation (Lewis et al., 2020; Guu et al., 2020) and $k$NN language models (Khandelwal et al., 2020).

**Prompt-based Augmentation**   Augmenting LLM prompts with retrieved documents has become a popular approach to incorporate a non-parametric datastore into an LLM-based pipeline (Lewis et al., 2020; Guu et al., 2020). For example, in a Retrieval Augmented Generation (RAG) setup (Lewis et al., 2020), the documents most relevant for a query are retrieved and added to the context in the prompt to an autoregressive language model.

**$k$NN Language Models**   In the $k$NN-LM architecture (Khandelwal et al., 2020), the language model produces an output distribution $p_{\mathrm{LM}}(y, x)$ *without* taking into account the retrieved documents. A separate distribution $p_{k\mathrm{NN}}(y|x, C)$ is computed based on the retrieved documents only, using $x$ as the criteria for document selection. Finally, a mixture of both distributions (parameterized by a hyperparameter $\gamma$) produces the final retrieval augmented output which is given by

$$p(y|x, C) = \gamma p_{k\mathrm{NN}}(y|x, C) + (1 - \gamma)p_{\mathrm{LM}}(y|x) . \tag{4}$$

While $p_{\mathrm{LM}}$ requires expensive LLM inference, the expression $p_{k\mathrm{NN}}$ can be computed efficiently by pre-computing a key-value store with a single forward pass over the datastore that is then queried at inference time.

**Label Propagation Wrapper**   Our approach can be integrated into both of the above architectures as a *wrapper* around the existing system. We assume that every document $c$ that can be retrieved has a label $\ell(c)$. In practice, if this is not the case, unlabeled documents can be assigned the most restrictive label (i.e. the top of the lattice). The process then proceeds as follows:

1. Given a prompt $x$, retrieve the context $C$ and run the model as usual on $(x, C)$ to obtain the original completion $y$.
2. Compute the pessimistic label $L$ of $C$, and run Algorithm 1 on $(C, L, x, y)$ to obtain a set $\Lambda$ of labels that are $\lambda$-similar to $L$.
3. Choose an appropriate $L' \in \Lambda$ using application-dependant criteria and run the model again on $(x, C_{|L'})$ to obtain a new completion $y'$.
4. Return the new completion $y'$ and the new label $L'$.

**Safety**   In practice, our algorithm is an AI-based heuristic that can make mistakes or be misled adversarially (e.g., failing to identify an influential document, or over-estimating the influence of a document). However, these mistakes do not affect the *safety* of the propagated labels. By rerunning the model on the new context $C_{|L'}$ (step 3) and returning only $y'$ (step 4), our approach guarantees that the returned completion depends only on documents at or below $L'$. This safety property holds even in the case of adversarial input documents (e.g., prompt injection) because it is enforced by the system, rather than the model.

### 3.5 Computational Cost

The label propagator wrapper based on prompt-based augmentation requires a number of LLM calls that, in the worst case, is exponential in the number of documents in the context. This occurs e.g., in a flat bounded lattice where all labels between $\bot$ and $\top$ are incomparable and when each subcontext has a different label.

However, the number of LLM queries is always bounded by the size of the lattice. When the lattice forms a totally ordered set (e.g. two-element lattices distinguishing between trusted vs untrusted or confidential vs

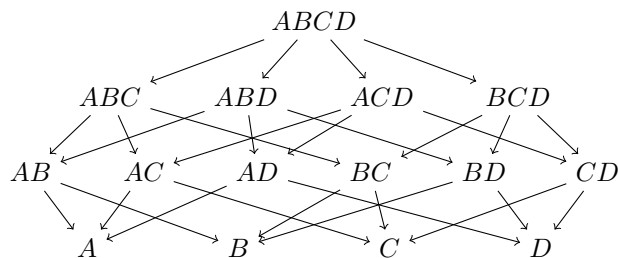

Figure 3: Illustration of the lattice for the synthetic key-value dataset with 4 documents. If a query requires multiple documents to produce the correct response, the corresponding label is the joint label of all the documents.

---

**Sample documents from the synthetic dataset**

*[Doc A]* The social security number and date of birth of person 1 is SSN00038242 and 26-10-1962.
*[Doc B]* The social security number of person 2 is SSN00092411.
*[Doc C]* The date of birth of person 2 is 18-08-1992.
*[Doc D]* The social security number and date of birth of person 2 is SSN00092411 and 18-08-1992.

---

**Sample QA pair from the synthetic dataset**

*[Question]* What are the social security numbers and date of birth of person 1, and person 2?
*[Answer]* The social security number and date of birth of person 1 is SSN00038242 and 26-10-1962, and person 2 is SSN00092411 and 18-08-1992.
*[Label]* $\{ABC, AD\}$

---

Figure 4: Sample documents and QA pairs from the synthetic key-value dataset. The question refers to the social security numbers and dates of birth of person 1 and 2. This information can be obtained by accessing documents $A$, $B$, and $C$ or $A$, and $D$, hence the resulting label of $\{ABC, AD\}$.

public data), Algorithm 1 stops as soon as it finds a subcontext whose utility drops below a $\lambda$ difference w.r.t. the utility of the full context. For richer lattices describing more fine-grained security policies, Algorithm 1 visits a small subset of the lattice in typical queries, either because only a few labels are represented in the context or because the utility drops below the tolerance $\lambda$. Therefore, computational costs are not a major concern for several fundamental lattices. We discuss several optimizations in Section 6.3 that reduce the cost of running the system. Liu et al. (2024a) also proposed tricks to further accelerate the computation of these context attributions.

To avoid worst-case costs, alternative algorithms or model architectures may be more suitable. For example, the $k$NN-based architecture we consider in this paper only requires one additional LLM call per query ($p_{\text{LM}}(y|x)$, in addition to the original query $p_{\text{LM}}(y|x, C)$).

## 4 Evaluation Setup

As this is a novel problem setup, we first define a robust evaluation scheme that we use to evaluate our approach and compare against an introspection baseline. Our goal is to understand the performance of our label propagator in correctly identifying and propagating minimal labels.

---

**Sample document from the news article dataset**

*[**High Integrity**]* Apple cuts prices on lower-end iPads, releases red iPhones
Apple is cutting prices on two iPad models and introducing red iPhones, but the company held back on updating its higher-end iPad Pro tablets.
A much-speculated 10.5-inch iPad Pro didn't materialize, nor did new versions of existing sizes in the Pro lineup, which is aimed at businesses and creative professionals. The new devices are mostly refreshes of existing models. Apple unveiled them through press releases Tuesday rather than a staged event, as it typically does for bigger product releases.

*[**Low Integrity**]* Apple cuts prices, on lower-end iPads, adds colors to the iPhone lineup
While the iPad Pro tablets didn't get an update, the two lower-end iPad models got a $100 price cut today, unveiled through a quiet press release rather than a large staged event. With fans clamoring for a greater variety of colors for their iPhones, Apple announced in the same release five fruit-inspired colors, hearkening to the flavors of the iMac G3 in 1998. The new colors, available starting next Tuesday, are Cherry (red), Lemon (yellow), Lime (green), Blueberry (blue), and Grape (purple).

---

**Sample QA pair from the news article dataset**

*[**Question**]* What are the new fruit-inspired colors for the iPhone lineup mentioned in the article about Apple cutting prices on lower-end iPads and adding colors to the iPhone lineup?
*[**Answer**]* The new fruit-inspired colors for the iPhone lineup mentioned in the article are Cherry (red), Lemon (yellow), Lime (green), Blueberry (blue), and Grape (purple).
*[**Label**]* {LoInt}

*[**Question**]* What is the new color introduced for the iPhone according to the article about Apple cutting prices on lower-end iPads and releasing red iPhones?
*[**Answer**]* According to the article, the new color introduced for the iPhone is red.
*[**Label**]* {HiInt}

---

Figure 5: Sample documents and QA pairs from the news article dataset. The first question can only be answered with access to the LoInt document whereas the second question can be also answered with the HiInt document.

## 4.1 Research Questions

The goal of the label propagator is to return all minimal $\lambda$-similar labels. As shown in Proposition 1, this is achieved under monotonicity assumptions that are typically not satisfied in practice. Therefore, our evaluation aims to quantify how closely the empirical results match this goal. Specifically, we aim to answer the following research questions:

RQ1: *How accurate is our label propagator in identifying the set of minimal labels?*
RQ2: *By how much does our label propagator improve over a naïve label propagator?*
RQ3: *How aligned is the regenerated output for the inferred label with the full-context output?*

Research questions RQ1 and RQ2 focus on the performance of Algorithm 1 directly. Research question RQ3 focuses on the quality of the final output of the end-to-end system introduced in Section 3.4.

## 4.2 Evaluation Metrics

We now describe the metrics that we use to answer the aforementioned research questions. Throughout, we use $\Lambda$ to denote the set of labels returned by Algorithm 1 and $\Lambda^\star$ as ground truth, i.e., the correct set of minimal labels.

**Exact match**   The exact match metric computes the average number of completely correct predictions over all questions. That is, for each question we count 1 if $\Lambda = \Lambda^\star$ and 0 otherwise. Exact match gives a direct answer to RQ1, but is very sensitive in that it equally penalizes any imprecision in the label propagator.

**Precision and Recall**   For a more fine-grained evaluation, we compute precision and recall between $\Lambda$ and $\Lambda^\star$ for every question, i.e., $|\Lambda \cap \Lambda^\star| / |\Lambda|$ and $|\Lambda \cap \Lambda^\star| / |\Lambda^\star|$, and average these over the entire dataset. Note that precision and recall simultaneously reach their maximum of 1 if and only if $\Lambda = \Lambda^\star$, i.e., we have an exact match.

**Label improvement**   When the lattice forms a total order (i.e., any two labels are comparable), both $\Lambda^\star$ and $\Lambda$ become singleton sets. Therefore, in this case, instead of reporting precision and recall over set outputs, we instead report *label improvement* and *missed labels*. We define label improvement as the number of cases in which our system improves the output label, as a fraction of the total number of cases where label improvement is possible. Missed labels, on the other hand, is defined as the number of cases our system misses a label from the context, as a fraction of the total number of cases where our system improves the label. Note that in the case of more than two labels, these metrics only consider perfect improvement (i.e., the output label exactly matches the ground truth label).

**Model output alignment**   The system for label propagation described in Section 3.4 regenerates the output based on the inferred label. While the definition of $\lambda$-similarity guarantees that under the subcontext of the inferred label the original output is almost as likely as in the full context, this does not necessarily guarantee that the regenerated output has good utility w.r.t. to other metrics. To quantify potential deviations, we measure the alignment of the different model outputs w.r.t. the target specified in the dataset i.e., $y^\star$ using ROUGE-L (Lin, 2004) metric, which is based on the Longest Common Subsequence (LCS) (Hunt & Szymanski, 1977) computation and has been commonly used to estimate translation quality in the past.

In particular, we report the average alignment of the full context output ROUGE-L$(y, y^\star)$, and the average alignment of the reduced context output after label propagation ROUGE-L$(y', y^\star)$. Furthermore, we report the average difference of the alignment between the reduced context output and full context output i.e., ROUGE-L$(y', y^\star)$ - ROUGE-L$(y, y^\star)$ in two different cases i.e., (i) when label improvement is possible, and (ii) when label improvement is not possible. The difference in output alignment should be nearly zero when label improvement is possible, and highly negative when such improvement is not possible.

### 4.3   Baseline

As a baseline for comparison, we use *introspection* in which the LLM itself is asked to determine which documents in the context were influential. This is inspired by the recent successful application of language models to generate relevant citations for their own outputs (Taylor et al., 2022; Gao et al., 2023). Note that the performance of the introspection technique depends significantly on the prompting technique used, and hence, should be considered as a lower-bound (Turpin et al., 2024).

### 4.4   Models

We focus on the Llama-2 model family (Touvron et al., 2023) including its 7B and 70B variants for evaluation. We use instruction-tuned versions of Llama-2 to ensure accurate response generation unless mentioned otherwise. For the $k$NN-LM implementation, we follow (Khandelwal et al., 2020) and use the model's penultimate layer representation of the last token (conditioned on all preceding tokens in the document) as the context representation for $k$NN search. Note that while we rely on Llama-2 model family (Touvron et al., 2023) for our experiments, our approach is also applicable to other open-source models, or even proprietary models accessible only via API (though client-side optimizations will be harder to implement).

### 4.5   Datasets

Since the problem of label propagation for LLMs has not been previously studied, there are no off-the-shelf datasets available for evaluation. We thus design three datasets to evaluate different aspects of our label

propagator. Each dataset consists of a set of documents and a set of corresponding question-answer pairs. The documents and answers all have labels. The goal of the label propagator is to identify the subsets of documents in a given context that are required for answering the question and have a label that is at least as permissive as the naïve label propagator.

We use the target response in the dataset as the completion $y$ for Algorithm 1, i.e., we do *not* rely on the model for generating the output unless mentioned otherwise. This is motivated by the fact that the target label computed in the dataset is only correct w.r.t. the target response. Therefore, an incorrect completion $y$ from the model would render the ground-truth label set $\Lambda^\star$ incorrect. We quantify the implications of this decision in Section 5.3, where we compare the difference in performance between the dataset target and the model-generated output.

**Synthetic key-value dataset**   We create a set of key-value pairs that contain hypothetical individuals' IDs as keys, and their Social Security Numbers (SSNs) and dates of birth (DoB) as values. We randomly split, distribute, and replicate the key-value pairs across multiple documents, and we attach a unique label to each document. We design the questions such that obtaining the correct answer requires identifying the different combinations of documents that contain all the necessary values. An example of the generated documents and QA pairs is shown in Figure 4. In this example, the question can be answered with access to documents $A$, $B$, and $C$ or with access to documents $A$ and $D$. Consequently, the label search should yield $\{ABC = A \sqcup B \sqcup C, AD = A \sqcup D\}$.

Since each document has a unique label, it corresponds to a node at the bottom of the lattice (similar to the example shown in Figure 3). Each set of documents (i.e., subset of the context) therefore corresponds to a unique element in the lattice. The documents and question-answer pairs are designed such that the question can be answered from different combinations of subsets in the context. Identifying all minimal labels corresponds to identifying all these subsets.

We generated a total of 128 documents (i.e., 128 labels) and 64 question-answer pairs. To ensure computational tractability, we set the context size to be 14 documents for each question. We ensure that all necessary documents are included in the context, thus emulating the case of a retriever component with perfect recall (all necessary documents are present) but lower precision (the context may contain irrelevant documents). This dataset represents a challenge in terms of the complexity and size of the search space because the full set of labels contains $2^{14}$ elements for each question. Since there is only a partial order in this lattice, we use the precision and recall metrics for evaluation.

**News article dataset**   Starting from an existing fake news dataset (Pérez-Rosas et al., 2018), we create pairs of high and low-integrity news articles that discuss similar topics to each other. Contrary to the previous dataset, we focus on a simpler lattice but more complex language in this case. Using GPT-4 (OpenAI, 2023), we generated QA pairs based on these articles, where some of the answers depended only on the LoInt document, others only on the HiInt document, and some on both documents. An example document pair and QA pair is shown in Figure 5. The dataset in total contains 240 document pairs of corresponding LoInt and HiInt documents, as well as 3465 QA pairs. Compared to a naïve label propagator, it is possible to improve the label on 647 of these QA pairs. Since there is a total order in the lattice, we report label improvement metrics for this dataset.

**LLM agent dataset**   Motivated by the recent emergence of LLM agents (Xi et al., 2023) and tool use in LLMs (Schick et al., 2023), we design an LLM agent dataset that focuses on tool use. Retrieval tools such as web or email search typically return multiple results of varying integrity levels. For example, in the case of web search, official documentation like `docs.python.org` might be deemed to be higher integrity than community curated sources such as `stackoverflow.com`. Similarly, emails from verified senders may be deemed to be higher integrity than those from unknown senders.

To demonstrate this, we create a small dataset of chat conversations between a user and an LLM-based assistant. In each conversation, the LLM calls one of three retrieval tools: web search, email search, or calendar search. Each piece of retrieved data carries a label $L \in \{\text{HiInt}, \text{LoInt}\}$ indicating high or low integrity respectively.

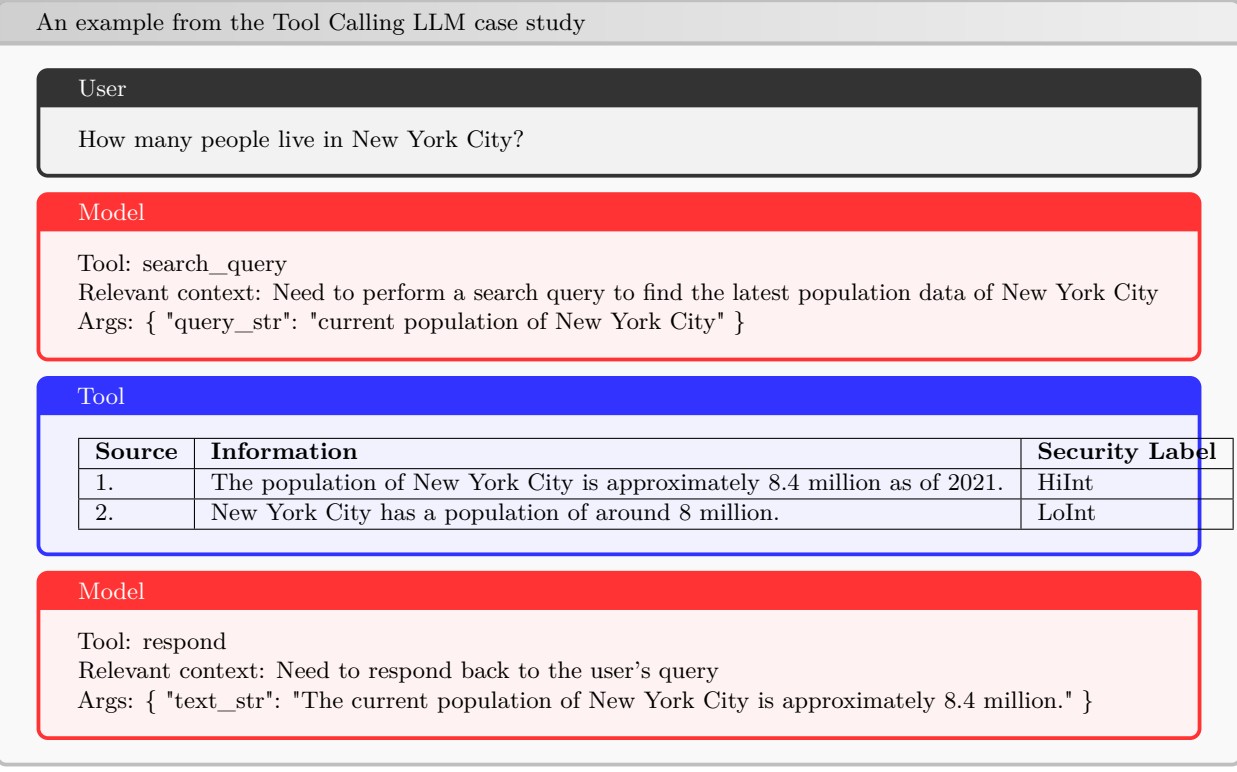

Figure 6: An example interaction from the tool calling LLM case study, where the user query triggers a web search. The items returned by the tool have different labels. The aim of our label propagator is to identify the label for the final model output. Since the high-integrity output is sufficient to answer the query in this case, the label propagator can upgrade the label of the output to high integrity instead of the naive baseline of propagating the low-integrity label.

In the example shown in Figure 6, the web search retrieves two documents that both contain sufficient information to answer the question. However, one of the documents carries a low-integrity label and would therefore force the output to be LoInt. The goal of the label propagator is to identify the influence of the high-integrity document and to assign HiInt to the output.

To cover a wider range of cases, we manually create 40 distinct chat conversations $x$ that involve tool calls. Some of these tools return a set of two documents $C$. We also generate a ground truth model output $y^\star$ and label $L^\star$. In 20 of the cases, the final label can be improved i.e. $L^\star \sqsubset \ell(C)$ due to redundant or irrelevant information within the retrieved LoInt documents. In the remaining 20 cases, the LoInt source is required to produce the output and thus the output label cannot be improved i.e. $L^\star = \ell(C)$. We additionally include two in-context examples that are specifically used to specify the output format for the model. Since there is a total order in the lattice, we report label improvement metrics.

## 5 Results

### 5.1 Synthetic key-value dataset

In this case, we assume a perfect recall retriever (all relevant documents are present), albeit lower precision (the rest of the documents out of the total limit of 14 are filled by adding irrelevant documents). For the introspection baseline, we use one-shot in-context learning (Wei et al., 2022) based on the first example in the dataset and use the remaining 63 questions for evaluation. Table 1 summarizes our results.

| Model | Label Prediction Method | Exact Match | Precision | Recall |
|---|---|---|---|---|
| Llama-2-Chat (7B) | Prompt-based | **85.94%** | $92.34 \pm 23.27$ % | $93.75 \pm 22.10$ % |
| | $k$NN-LM | 53.12% | $64.58 \pm 45.79$ % | $65.10 \pm 45.33$ % |
| | Introspection | 1.59% | $3.25 \pm 17.53$ % | $3.97 \pm 18.48$ % |
| Llama-2-Chat (70B) | Prompt-based | **85.94%** | $94.17 \pm 21.84$ % | $92.06 \pm 23.45$ % |
| | $k$NN-LM | 57.81% | $69.53 \pm 45.60$ % | $64.45 \pm 44.74$ % |
| | Introspection | 12.70% | $15.07 \pm 34.94$ % | $15.87 \pm 35.44$ % |

Table 1: Results on the synthetic key-value dataset assuming a perfect recall retriever, which always retrieves the relevant items while also retrieving some irrelevant ones, with person ID as the key, and person SSN and DoB as values. $k$NN-LM prediction uses $\gamma = 0.5$. We report the mean and one standard deviation for macro-averaged precision and recall.

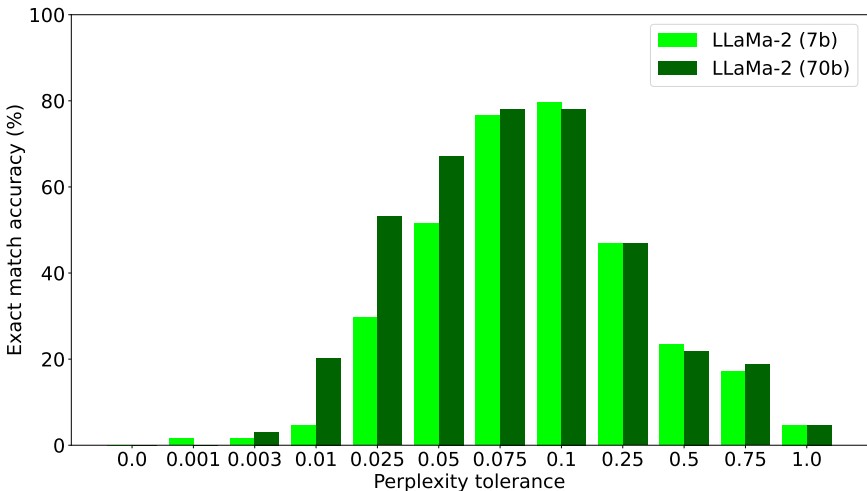

Figure 7: Hyperparameter grid search on perplexity tolerance $\lambda$ for the 7B and 70B models on the synthetic key-value dataset. See Figure 9 for the full grid search over both Shapley threshold and perplexity tolerance.

The prompt-based propagator achieves an exact match accuracy of over 85% and precision and recall of over 90%. That is, in more than 85% of the cases, the label search can identify the correct influential subcontexts out of a set of 16k possibilities without error. Furthermore, the prompt based label search significantly outperforms both the $k$NN-LM-based label search as well as the introspection baseline.

Table 1 also compares the exact match accuracy for two different model sizes 7B and 70B parameters. The model size does not strongly correlate with the performance of the prompt based label propagator. Consequently, this enables the use of significantly smaller models for effective label propagation on the outputs of larger models.

On the other hand, introspection performance improves significantly with the model scale. This result is consistent with the improved in-context learning ability observed in larger models (Wei et al., 2022). Therefore, only the largest and most powerful models can be effectively used for introspection.

**Sensitivity of hyperparameters** The perplexity tolerance $\lambda$ measures the acceptable loss in model utility when removing a subset of documents. The choice of $\lambda$ allows to trade-off model utility and the restrictiveness of the inferred label.

We illustrate the role of the perplexity tolerance $\lambda$ on the resulting change in the exact match performance for our prompt based label propagator in Figure 7. The figure indicates that the model performance is very sensitive to the choice of perplexity tolerance i.e., either very high or very low values significantly hamper

| Retriever Type | Model | Exact Match | Label Improvement | Missed Labels |
|---|---|---|---|---|
| Perfect Retriever | Llama-2-Chat (7B) | 74.69% | 56.72% | 22.73% |
| | Llama-2-Chat (70B) | 73.45% | 39.26% | 18.59% |
| Realistic Retriever | Llama-2-Chat (7B) | 63.78% | 49.77% | 30.30% |
| | Llama-2-Chat (70B) | 64.50% | 39.88% | 26.08% |

Table 2: Results on the news article dataset with prompt-based label search. The perfect retriever row assumes that the retriever always retrieves the relevant items while also retrieving some irrelevant ones randomly. The realistic retriever uses a cosine similarity-based nearest-neighbour search and may not retrieve all relevant items, thereby reducing the label improvement metric of even a perfect label search.

model performance. Furthermore, the optimal hyperparameters are consistent across model scales. However, larger models, being better at language modeling, outperform their smaller counterparts at lower values of $\lambda$.

Although the influence-based label propagator performs very well on its own, we have found that pruning irrelevant labels can elevate the performance further. This is particularly the case for large lattices with a partial order. We introduce a Shapley-value-based heuristic pruning technique in Appendix A that can marginally improve the performance from an exact match accuracy of 81% to 86%. The results reported in Table 1 include this Shapley-value-based heuristic.

### 5.2 News article dataset

Moving towards a more realistic setup, we include an additional retriever component for the news article dataset that first retrieves relevant articles (either low or high integrity) from the dataset before response generation. In order to correctly understand the impact of the retriever, we compare a perfect retriever that is able to retrieve all relevant documents (similar to the synthetic key-value dataset) with a realistic retriever using cosine similarity in the embedding space computed by *BGE-Large-EN* (Xiao et al., 2023b).

Table 2 summarizes the results of the label propagator on the news article dataset. For this dataset, the label propagator achieves slightly worse performance on all metrics in comparison to the synthetic key-value dataset due to the lower quality of the dataset (automated generation of QA pairs by GPT-4 (OpenAI, 2023)). Furthermore, we see a reduction in the exact match accuracy of about 10% when using a realistic retriever in contrast to a perfect retriever. This is due to the retriever sometimes failing to retrieve the relevant pieces of information, ultimately leading to a mismatch with the ground-truth label.

Due to the presence of total order on the labels, we report label improvement and missed labels for this dataset. Even when a perfect retriever is used, a naïve label propagator would be overly conservative in 647 out of 3465 examples where label improvement is possible. In contrast, our influence-based label propagator is able to assign the correct label in $\sim 57\%$ of these cases. On the other hand, our label propagator suggested a more permissive label but missed a label from the ground-truth in $\sim 23\%$ of the cases. However, as explained in Section 3.4, the safety property still holds in these cases because the system regenerates the output using only the articles with the more permissive label (although this means that the regenerated output may differ from the original output, as we quantify in the next section).

**Sensitivity of hyperparameters** We highlight the sensitivity to changes in the perplexity threshold $\lambda$ on the news article dataset in Figure 8. Similar to the synthetic key-value dataset, we see similarities across the two model scales, and a higher tolerance of the larger model to lower values of $\lambda$. However, we see a significant rise in optimal perplexity tolerance in contrast to the synthetic dataset due to the higher difficulty of the responses.

### 5.3 LLM agent dataset

In the LLM agent dataset, we take a step further by also computing the misalignment introduced by regenerating the output conditioned on the updated context (when label improvement is possible). The

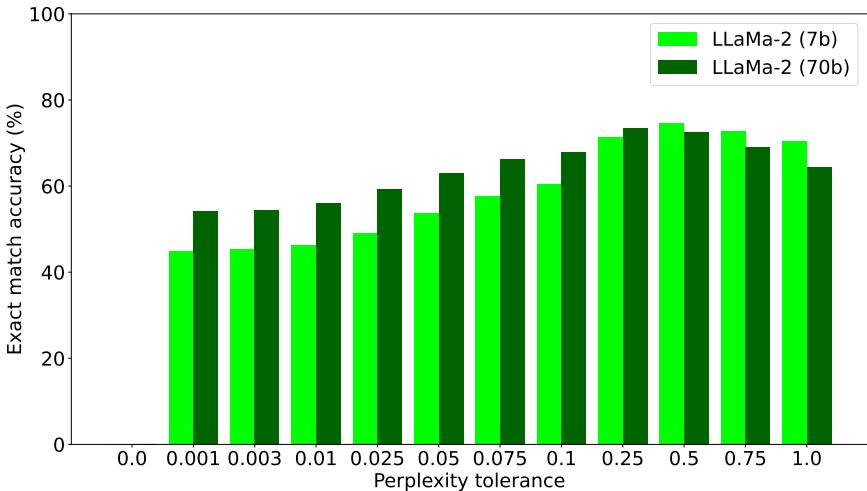

Figure 8: Hyperparameter grid search on perplexity tolerance $\lambda$ for the 7B and 70B models on the news article dataset with a perfect retriever. See Figure 10 for the full grid search over both Shapley threshold and perplexity tolerance.

| Model | Label Prediction Method | Utility Target | Label Improvement | Missed Labels |
|-------|------------------------|----------------|-------------------|---------------|
| | Prompt-based | Model generated $y$ | 85.0% | 5.56% |
| Llama-2 (7B) | Introspection | - | 45.0% | 0.0% |
| | Prompt-based | Ground truth $y^\star$ | 100.0% | 0.0% |
| | Prompt-based | Model generated $y$ | 95.0% | 9.53% |
| Llama-2 (70B) | Introspection | - | 80.0% | 11.11% |
| | Prompt-based | Ground truth $y^\star$ | 100.0% | 0.0% |

Table 3: Results on the LLM agent dataset. The utility target column indicates which output $y$ we are using to compute our target utility $U(p_{\mathrm{LM}}(y|x, C))$ in equation 1. In practice and in the absence of ground truth, the utility target is the model-generated output $y$, but we also compare with the ground truth $y^\star$ to understand the maximum possible utility assuming a perfect LLM. The difference in the numbers between the cases indicates that all the errors in the case of model-generated output are artifacts of the model generating a suboptimal response which causes incorrect propagation of labels.

results are presented in Table 3. We fix a threshold of $\lambda = 0.2$ for this experiment. We use the base model without instruction tuning in this case due to the use of a custom chat format.

First, we focus on the 20 cases where label improvement is possible i.e., $L^\star \sqsubset \ell(C)$. We find that the label propagator improves the label in at least 17 of these cases.

In at most 2 of the cases, the LP is overly optimistic and returns a more permissive label $L'$ than the ground truth label $L^\star$. Again, the safety property still holds in these cases because the system regenerates the output using the reduced context $C_{|L'}$. This regeneration step introduces the possibility that the new output $y'$ differs significantly compared to the output $y$ from the full context.

To quantify this, we measure the alignment between $y$ and $y^\star$ as well as $y'$ and $y^\star$ using the ROUGE-L F-score (see Table 4). We obtain a ROUGE-L F-score of at least 0.85, suggesting that conditioning on the reduced context leads to a similar model response. We see a negligible difference in alignment between the full context output and the reduced context output when label improvement is possible, but observe a significant drop when such an upgrade is not possible.

| Model | ROUGE-L$(y, y^\star)$ | ROUGE-L$(y', y^\star)$ | ROUGE-L$(y', y^\star)-$ ROUGE-L$(y, y^\star)$ | |
| --- | --- | --- | --- | --- |
| | | | Improvement possible | Not possible |
| Llama-2 (7B) | 0.77 | 0.85 | 0.0051 | -0.34 |
| Llama-2 (70B) | 0.82 | 0.90 | 0.025 | -0.31 |

Table 4: Model output alignment with the dataset ground truth computed using ROUGE-L F-score on the LLM agent dataset. Utility computation is only applicable to the prompt-based label propagator. ROUGE-L$(y, y^\star)$ refers to the alignment of the full context output (no regeneration required), while ROUGE-L$(y', y^\star)$ represents the alignment of the regenerated response after label improvement. Interestingly, we see that the regenerated response is more aligned with the ground truth response suggesting that in some cases label propagation can improve utility. ROUGE-L$(y', y^\star)-$ ROUGE-L$(y, y^\star)$ compares the difference in alignment between the full context and the subcontext of the inferred label. When label improvement is possible, the difference in alignment is close to nil indicating that the regenerated response $y'$ is highly aligned with the ground truth response $y^\star$. Otherwise, the difference in alignment is highly negative, indicating a significant drop in model alignment with $y^\star$.

In a robust retrieval augmentation setup, adding additional documents to the context should not negatively influence the model's ability to answer a query since the model is free to ignore irrelevant inputs (Shi et al., 2023). Thus, we expect that the ground truth completion $y^\star$ is always similar to the completion given the full context $y$. However, in almost all label propagation errors we observe, we find that the model is not able to answer the query with the full context. When controlling the model generation by using the ground truth target in the dataset instead of using the model-generated output, we see that our label propagator achieves perfect precision and recall.

Since the lattice is particularly simple in this case, we observe strong performance from the introspection baseline in contrast to the more complex lattice in the synthetic key-value dataset. Furthermore, similar to the previous datasets, we see significant improvement in the introspection performance with increasing model scale, highlighting that introspection might be particularly well-suited for larger models.

## 6 Analysis and Discussion

In this section, we highlight the main findings of our work and discuss the implications of our results when used in a real-world setting.

**RQ1**. We find that our prompt-based label propagator can find the exact set of minimal labels in 86% of the cases, for a large lattice of 16k possible labels.

**RQ2**. We evaluate the label propagator on a smaller lattice with a total order, allowing us to compare labels directly and quantify the label improvement. In this case, our label propagator improves the label in 56% of the cases for the news article dataset and 85% of the cases for the LLM agent dataset.

**RQ3**. We showcase the label propagator in a real-world use case in a tool calling LLM agent setup where the propagated label is used to determine whether a sensitive tool call is allowed. We find that the prompt-based label propagator is able to improve the label in more than 85% of the cases while the output of the LLM agent remains the same as measured by the difference between the reduced context output alignment and the full context output alignment.

**Comparison to baselines** The introspection-based label propagator achieves a noticeable improvement in performance when using a 70B parameter model compared to a 7B model. Furthermore, despite its simplicity and computational convenience, the introspection-based label propagator constitutes a strong baseline when considering simple lattices. We find a similar performance of the prompt-based label propagator for both model sizes suggesting that our approach could be implemented on smaller models, saving computational resources.

### 6.1 Use-cases

Algorithm 1 identifies sub-contexts with labels that are more permissive than that of the full context. However, whenever the desired output label $L$ can be determined *up-front*, it is possible to skip the search over $\lambda$-similar sub-contexts and restrict the retrieval component to documents at or below $L$.

Our approach reveals its true benefits when the use of the generated content is *not* yet determined after the initial retrieval step. In such cases, having a too-restrictive label comes at the cost of limiting future uses of the generated output. For example, semantic caches (Bang, 2023) can be extended to store answers along with a sensitivity label, where more permissive labels facilitate broader reuse of the generated content. Likewise, Emails are often forwarded beyond their initial set of recipients, which is facilitated by using permissive labels.

Our approach naturally allows users to *endorse* information. Let's consider a setup as illustrated in Figure 1. Initially, there might only be an untrusted source answering the initial query. The LP system would correctly output an untrusted label to the potentially dangerous command. However, after a trusted authority (e.g., a university department IT admin) confirms the suggestion in the untrusted source, the LP recognizes that both sources are similarly influential and assigns a trusted label to the output. Therefore, by quoting or repeating untrusted information a trusted source can *endorse* information.

### 6.2 Limitations

Our label propagator is able to improve on the baseline label by more than 50% and 85% in two realistic datasets. However, this improvement comes with the cost of an increased number of LLM calls (discussed in Section 3.5).

In the presence of an adversary, our label propagator assigns the lowest integrity label by design in order to avoid any harmful side effects such as cross-prompt injection (Liu et al., 2023). However, this can potentially lead to a degradation of service attack as the adversary can add unreliable distracting information that makes it impossible for the system to improve the label, reducing the downstream utility of the model output.

### 6.3 Extensions

**Other applications**  Our main focus has been on predicting the least conservative label while ensuring that utility is not compromised beyond a certain threshold $\lambda$ (Definition 1). However, our proposal is flexible enough to accommodate various use cases. For instance, it can be reversed to determine the utility for a specific target label. Alternatively, a more complex use case would be to make the system dynamic with respect to $\lambda$, which means the system is able to compromise more utility to achieve a better or less conservative label.

**Efficiency improvements to Algorithm 1**  Section 3.5 discusses the computational cost incurred to execute our label propagator in Algorithm 1. Despite an increase in the number of LLM calls for label propagation, the cost incurred per call can be drastically reduced utilizing common inference optimizations implemented in model serving backends (Zheng et al., 2023; Kwon et al., 2023). For instance, compute-bound prompt processing in Transformers can be amortized across calls sharing a common prompt prefix by reusing a KV cache. For a totally ordered lattice, appending retrieved documents at the end of the prompt sorted by their labels will result in no additional prompt processing costs in subsequent LLM calls because their prompts are obtained by *peeling off* documents at the end of the prompt.

We can also optimize the decoding phase of an LLM call by keeping a running calculation of the cumulative likelihood of the tokens decoded and stop decoding when the utility drops below a difference $\lambda$ of the utility of the full context. Liu et al. (2024a) also explored tricks to improve the efficiency of context attribution in LLMs.

**Advanced influence estimation**  The label propagation mechanism developed in Section 3 is based on estimating the influence of context elements on the model's predictions. There are several alternatives

techniques, such as Shapley value estimation, that can be used as drop-in replacements. See Brophy et al. (2023) for an overview of influence estimation techniques and (Nguyen & Wong, 2023; Cohen-Wang et al., 2024) for examples of their application to LLM contexts.

Another approach is to leverage the attention weights for influence estimation. Phenomena such as attention sinks (Xiao et al., 2023a) or lost-in-the-middle (Liu et al., 2024b) makes identification based on simple attention scores difficult.

# 7 Related Work

**Information-flow analysis**   Information-flow analysis and the use of flow labels has a long-standing history, see e.g., Denning (1976), with an early survey of language-based approaches (Sabelfeld & Myers, 2003). The operations considered in many program analysis frameworks (e.g. Volpano et al. (1996); Sabelfeld & Myers (2003)) have well-defined semantics, and the rules for propagating the flow labels of each operator can be hard-coded. The situation is different when using ML models such as Transformers: due to their attention mechanism and autoregressive decoding, their output is tainted by all inputs. Moreover, their behavior is only implicitly defined as the solution of a loss-minimization problem. In our approach, we propagate labels through ML components under the constraint that the model's loss does not increase.

The literature distinguishes between static and dynamic approaches to information flow control (Sabelfeld & Myers, 2003; Buiras et al., 2014; Mardziel et al., 2011). The key difference is that dynamic approaches track the flow during program execution, whereas static approaches reason collectively about *all* possible executions without actually executing the program. Our approach relies on analyzing *one* specific context and is hence fundamentally dynamic.

Secure multi-execution (SME) (Devriese & Piessens, 2010; Rafnsson & Sabelfeld, 2016) is an approach to dynamic information-flow analysis where programs are executed once per security level. The idea is that an execution that produces output for level $L$ receives data only from levels $L' \sqsubseteq L$; and dummy values from levels $L'' \sqsubseteq L$. Programs that are multi-executed are secure by design, and correct if the original program was secure. Our approach is related to SME in that we also call the LLM once per security level, and we propagate the most permissive label for which the original output is still "correct", in the sense that the drop in utility is bounded by $\lambda$.

**Information-flow analysis in AI applications**   Several emerging approaches rely on information-flow tracking to obtain deterministic security guarantees for LLM-based applications, even if the underlying models are vulnerable to attacks: Wutschitz et al. (2023) study how to leverage existing metadata in datastores used for retrieval augmentation to enforce confidentiality guarantees at inference time. Wu et al. (2024) propose a system that uses information-flow tracking to prevent LLM-based applications from being compromised by untrusted information. Balunovic et al. (2024) develop a monitor and policy language that can check execution traces of LLM applications for violations of different safety and information-flow properties. An emergent line of work uses data and information-flow tracking to enforce deterministic security guarantees for the tools called by AI agents (Costa et al., 2025; Debenedetti et al., 2025; Zhong et al., 2025). In contrast to these approaches, our work aims to dynamically and soundly propagate labels *through* LLM calls, which can be used to increase the permissiveness of system-level information-flow tracking.

Related to the introspection baseline, Mireshghallah et al. (2024) and Ghalebikesabi et al. (2024) use Contextual Integrity theory to explore how well LLMs can be relied upon to discern which information is appropriate to use in a given context. Finally, Wallace et al. (2024) train LLMs to distinguish between instructions in their context totally ordered in a hierarchy, teaching them to prioritize higher-privileged instructions over instructions that appear lower in the hierarchy.

# 8 Conclusion

We presented a permissive approach to propagating information-flow labels of documents retrieved in RAG systems. The key idea is to propagate only the labels of those documents that are actually used for generating

the model's output. We show that our approach is practical in terms of performance and infers more permissive labels than an introspection baseline. Unlike commonly used introspection-based methods, our approach can satisfy a hard safety guarantee.

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

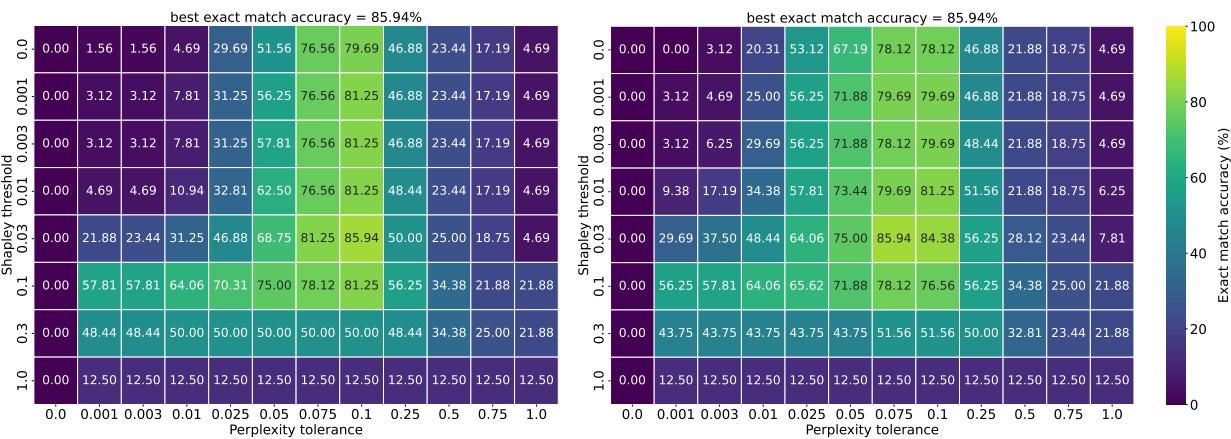

Figure 9: Hyperparameter grid search on Shapley threshold and perplexity tolerance for the 7B (left) and 70B (right) models on the synthetic key-value dataset.

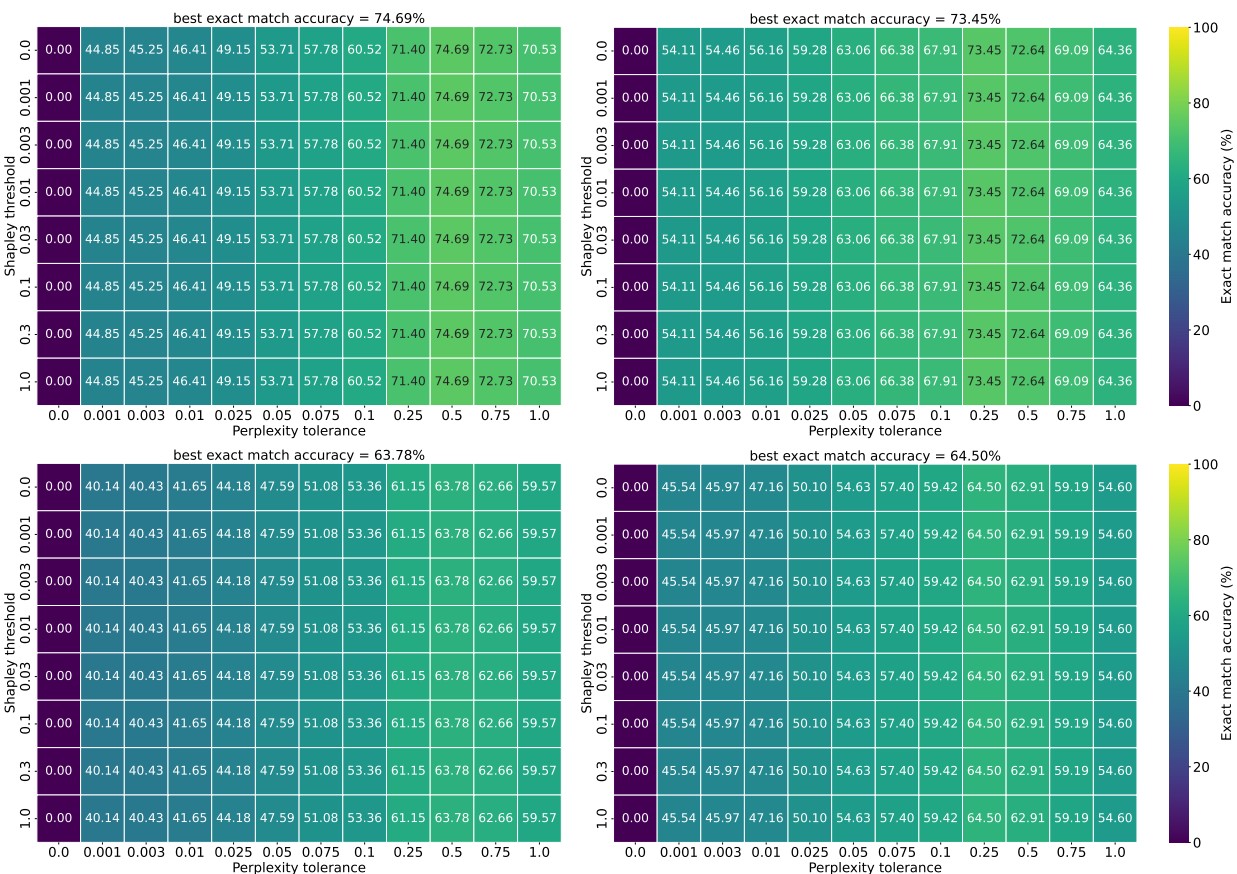

Figure 10: Hyperparameter grid search on Shapley threshold and perplexity tolerance for the 7B (left) and 70B (right) model on the news article dataset with a perfect retriever (first row) and realistic retriever (second row).

## A Improving performance on fine-grained lattices

Algorithm 1 relies only on $\lambda$ to produce the minimal set of labels. However, in the cases where documents share similarities and provide important information regarding the model output such as the format (which

is the case for our synthetic key-value dataset 4.5), the overall contribution of irrelevant documents goes up. Therefore, additional irrelevant documents are introduced in the predicted label set due to being $\lambda$-similar.

In order to cover these cases when dealing with a complex lattice and a large number of relations, we propose a simple heuristic based on Shapley values that have been commonly used in the past Shapley (1953); Nguyen & Wong (2023). In particular, we compute the Shapley value for each of the different labels in the lattice and filter out any label combinations where labels below a particular threshold are present.

The Shapley value defines the marginal contribution of any label $L$ by computing the average difference in outcomes when a particular label is present and absent. We compute the label Shapley value via perplexity, which is computed as the average drop in perplexity when a particular label is included. This provides a notion of the importance of each label. This now introduces an additional hyperparameter i.e., the Shapley value threshold.

We visualize the results for the 2D grid search by considering both perplexity tolerance $\lambda$ as well as Shapley threshold on the synthetic key-value dataset in Figure 9, and the news article dataset in Figure 10.

The results indicate that on simpler lattices such as in the case of news articles, we see almost no impact due to the Shapley value threshold. On the other hand, for complex lattices, such as in the case of our synthetic key-value dataset, we observe 6% absolute improvement in the exact-match accuracy, highlighting the utility of this heuristic in such cases.

