# OpenReview forum: "Permissive Information-Flow Analysis for Large Language Models"
_TMLR — Accepted by TMLR_

### Review · Reviewer_mg3P · 2025-07-14

**Summary Of Contributions:**

The paper introduces influence-based label propagation (LP) for retrieval-augmented LLM systems. Instead of conservatively assigning the join of all input labels to a model’s output, the proposed LP only propagates the labels of influential context documents, determined by bounding the drop in utility (perplexity) within λ. The authors study two concrete instantiations: (i) prompt-based retrieval augmentation in which retrieved passages are concatenated into the prompt (typical RAG setup), and (ii) kNN-LM, where an external datastore contributes a mixture distribution to the next-token probabilities. Across three datasets (synthetic key-value, news, and an LLM-agent tool-use corpus) the prompt-based LP achieves exact-match accuracies up to 86% and improves or preserves the output label in > 85% of agent scenarios 2025. The paper provides a formal problem statement, correctness guarantees under monotonicity, and an algorithm whose search complexity is bounded by the lattice size.

**Audience:**

Yes

**Claims And Evidence:**

Yes

**Requested Changes:**

- Connect to existing threat-model frameworks:
	- Please discuss and explicitly cite Operationalizing a Threat Model for Red-Teaming Large Language Models (LLMs) by Verma et al., 2024 especially in the context of Infusion Attacks presented in their work, exploring integration possibilities with structured adversarial capabilities and evaluation frameworks presented therein.
		- https://arxiv.org/pdf/2407.14937
- Provide practical guidance on setting λ:
	- Recommend an analytic approach or adaptive heuristic for setting λ in real-world deployments to reduce the tuning burden.
- Clarify and analyze kNN-LM shortcomings:
	- Conduct and include an ablation study on the γ hyperparameter and provide qualitative insights on factors contributing to poor kNN-LM performance.
- Include cost and latency evaluations:
	- Add concrete runtime metrics (wall-clock time, token count) comparisons between LP, introspection, and baseline methods to quantify practical trade-offs.
- Evaluate robustness to incorrect labeling:
	- Include experiments quantifying the impact on accuracy and safety guarantees when document labels are partially incorrect or missing.
- Clarify fallback behavior:
	- Explicitly describe the system's response strategy when no sub-context meets the λ-similarity condition to clearly communicate the safety fallback mechanisms.
Overall, the paper presents valuable insights and methods addressing key security and usability concerns for LLM deployments. Addressing the suggested revisions will strengthen its practical applicability, robustness, and connection to broader security research.

**Strengths And Weaknesses:**

**Strengths**

- Clear conceptual framework:
	- Clearly defines λ-similarity and minimal influence labels, ensuring practical interpretability.
	- Formal correctness and safety guarantees enhance trustworthiness.
- Practical applicability:
	- Directly addresses "label creep," an important real-world security and privacy challenge.
	- Enables practical deployment scenarios, particularly where data sources have varying trust or confidentiality levels.
- Comprehensive empirical evaluation:
	- Novel setting, robust experiments covering multiple scenarios and lattice complexities.
	- Demonstrates consistent performance improvements over baselines, particularly introspection methods.
- Algorithmic efficiency:
	- Provides efficient heuristic methods (e.g., Shapley pruning) for large search spaces, maintaining performance without compromising safety.
- Security focus:
	- Effectively addresses indirect prompt-injection attacks by limiting label propagation to necessary contexts only.


**Weaknesses**

- Deployment complexity and cost:
	- Reliance on repeated model execution (especially in commercial APIs) might increase costs and latency significantly.
- Dependence on accurate labeling:
	- The approach assumes accurate and complete labeling of all retrieved documents, which might not always be realistic in practice.
- Limited insights on kNN-LM performance:
	- Notably weaker results in kNN-LM implementation lack detailed analysis and explanation.
- Hyperparameter sensitivity:
	- Narrow optimal range for the hyperparameter λ; insufficient guidance on practical setting strategies.
- Potential denial-of-service vulnerabilities:
	- Acknowledged but not thoroughly explored possibility of denial-of-service via injection of low-utility, low-integrity context.

---

> ### Author Response · Authors · 2025-08-07
> **Response to reviewer mg3P**
>
> We are very thankful to the reviewer for the useful and constructive feedback. We address the points raised by the reviewer below:
>
>
> ### Requested changes
>
>
> > Connect to existing threat-model frameworks
>
> Thank you for the reference. We have added the suggested citation to the paper.
>
>
> > Provide practical guidance on setting λ:
>
> A grid search might be the most appropriate way to tune this hyperparameter, as suggested in the paper, since λ is scenario-specific, and system designers would need to evaluate different λ values on their specific models/datasets. Figure 7 shows one strategy for selecting λ (i.e., maximize exact matches), but other strategies are equally valid (e.g., keeping outputs similar to the original by lowering λ).
>
>
> > Clarify and analyze kNN-LM shortcomings
>
> Thank you for the suggestion to clarify the shortcomings of kNN-LM and include an ablation study. We agree that a more detailed analysis is beneficial.
>
> We selected kNN-LM as a baseline primarily because its architecture facilitates efficient label propagation, which was useful for our sub-context identification task. However, it has fundamental limitations as a retrieval-augmented generation method. The primary shortcoming is its architectural inability to leverage the in-context learning capabilities of modern large language models (LLMs). The kNN-LM mechanism interpolates the LLM's output distribution with a distribution from the datastore via a hyperparameter, γ. This is a shallow form of integration that occurs after the main model has processed the input.
>
> This approach contrasts sharply with modern prompt-based retrieval, where retrieved documents are inserted directly into the model's context window. This allows the LLM's attention mechanism to deeply condition the generation process on the retrieved information, which is a significantly more powerful paradigm.
>
> Regarding the ablation study, our initial tuning revealed critical issues. We found that improving performance was challenging, as the model was prone to collapse due to the convex combination. Furthermore, we observed that the optimal choice of γ was highly dataset-dependent, indicating a lack of robustness. These findings suggest the performance bottleneck of kNN-LM is not simply a matter of hyperparameter tuning, but a fundamental limitation of its post-processing integration architecture.
>
> Based on these insights, we concluded that a detailed ablation on γ would not yield new understanding and chose to focus on kNN-LM as a baseline for its structural properties rather than as a competitive method for retrieval-augmented generation.
>
>
> > Include cost and latency evaluations:
>
> While we give rough estimates regarding cost and latency of evaluation, we intentionally avoided concrete comparison as we acknowledge that our implementations aren’t highly optimized. Comparing numbers without highly optimized implementations might give a false sense of speedup obtained with different approaches. We consider this an important and practically useful piece for future work.
>
>
> > Evaluate robustness to incorrect labeling:
>
> Our threat-model assumes that the assigned labels are correct. In case the labels are incorrect, an incorrect label would be propagated, compromising the reliability of the full system (despite the propagated labels being correct w.r.t. the input as guaranteed by our system). Dealing with cases where the labels themselves can be noisy is beyond the scope of the current work.
>
>
> > Clarify fallback behavior
>
> Thank you for highlighting this. In the case where none of the labels are λ-equivalent to the full context set, we assume that each element in the context set is important for that prediction. Hence, we propagate the worst label in the entire context which is the best prediction we can make as we assume that the model made use of the information from each of these different sources. We attempted to clarify this in the paper.

---

### Review · Reviewer_pCjM · 2025-07-14

**Summary Of Contributions:**

This paper addresses the problem of information-flow label propagation in LLM-based systems, where LLMs ingest and generate outputs conditioned on data of varying integrity. The key contribution is the proposal of a more permissive, influence-based label propagation algorithm that propagates only the labels of context documents actually required to produce the output, mitigating label creep from overly conservative standard approaches.

**Audience:**

Yes

**Claims And Evidence:**

Yes

**Requested Changes:**

1. It would be better to discuss how to tune $\lambda$ given its significant sensitivity.

2. It would be better to include experiments and analysis on mis-labeled situations.

**Strengths And Weaknesses:**

**Strengths**

1. The motivation is clear: The problem of label creep in naive label propagation for LLMs is convincingly articulated.

2. The influence-based label propagator is formalized with clear mathematical definitions.

3. The paper evaluates the method on three datasets to test label propagation in practical scenarios (synthetic, news, agent), and in most cases the method is effective.

**Weaknesses**

1. The method assumes all context documents are pre-labeled with correct integrity/confidentiality. In many real-world LLM workflows (such as web search, multi-agent collaboration, tool outputs), this assumption can break.

2. The introspection baseline is a bit weak for smaller models, which may make the improvement margins appear large.

3. The datasets, while thoughtfully constructed, are mostly synthetic or manipulated, thus, generalizability to more diverse, noisy, or large-scale real-world deployments is not verified.

---

> ### Author Response · Authors · 2025-08-07
> **Response to reviewer pCjM**
>
> We are very thankful to the reviewer for the useful and constructive feedback. We are glad to hear that the reviewer found the motivation and influence-based label propagator to be mathematically clean. We address the points raised by the reviewer below:
>
>
> > The method assumes all context documents are pre-labeled with correct integrity/confidentiality. In many real-world LLM workflows (such as web search, multi-agent collaboration, tool outputs), this assumption can break.
>
> We agree that our work assumes all context documents are pre-labeled. However, for all such applications, it is possible to define the appropriate reliability labels (we assume labels are always correct; please see our response to your last comment, where we highlight that such a system can be compromised under incorrect labels). As an example, web search is based on computing the popularity of a web page (i.e., the PageRank algorithm). Similarly, it is possible to define labels for custom applications when integrating them in an agentic workflow (such as tool-use). Therefore, while seemingly difficult, we consider labels to be obtainable for the given application domain. We elaborate on the issue of “obtaining labels in the wild” in the paper.
>
>
> > The introspection baseline is a bit weak for smaller models, which may make the improvement margins appear large.
>
> We think this is an interesting yet subtle point that we wanted to highlight in the paper. Introspection inherently relies on a powerful model with strong in-context learning abilities. Based on the scaling laws, it is already clear by now that scaling models is a reliable way to enhance model capabilities. Hence, it is conceivable that smaller models are much worse in terms of their in-context learning abilities and, subsequently, introspection compared to larger and more powerful models.
>
> Given a sufficiently powerful model and a simple lattice, introspection might be a feasible solution for label propagation in many cases, as highlighted in the paper. Similarly, even powerful models could fail at introspection with a complex lattice. We specifically wanted to highlight the fact that small models are bad at introspection in the paper, rather than highlighting the improvement margin as the reviewer pointed out.
>
> For the simpler lattice in the LLM-agents dataset, we highlight the gain with the use of newer models, which are also significantly better at introspection, even at the smaller model scales (please see the response to reviewer SC3C).
>
>
> > It would be better to discuss how to tune λ given its significant sensitivity.
>
> A grid search might be the most appropriate way to tune this hyperparameter, as suggested in the paper, since λ is scenario-specific, and system designers would need to evaluate different λ values on their specific models/datasets. Figure 7 shows one strategy for selecting λ (i.e., maximize exact matches), but other strategies are equally valid (e.g., keeping outputs similar to the original by lowering λ).
>
>
> > It would be better to include experiments and analysis on mis-labeled situations.
>
> Our threat-model assumes that the assigned labels are correct. In case the labels are incorrect, an incorrect label would be propagated, compromising the reliability of the full system (despite the propagated labels being correct w.r.t. the input as guaranteed by our system). Dealing with cases where the labels themselves can be noisy is beyond the scope of the current work.

---

### Review · Reviewer_SC3C · 2025-07-25

**Summary Of Contributions:**

The paper proposes a technique for improved label attribution for LLM generation given multiple sources in the context. The proposed algorithm propagates the labels of the samples that were influential in generating the model output and to eliminate the labels of unnecessary inputs.

**Audience:**

Yes

**Claims And Evidence:**

Yes

**Requested Changes:**

1) Rewrite the abstract and the introduction of the paper for giving better motivation and overview of the problem.
2) Perform experiments with 3-4 more open-source models such as various Qwen and Llama-3 models.
3) Increase the complexity of the dataset and present individual error analysis for subset of the problems for both the baseline and the proposed technique.

**Strengths And Weaknesses:**

## Strengths:
- The approach is simple and the paper discusses both correctness and the computational cost.
- The technical part of the paper is well written and formalized to a reasonable extent.
- The paper attempts to solve a new problem and thus curates various synthetic and manually generated datasets for the problem.

## Weakness:

**Writing** (Mainly in abstract and intro):

Overall the abstract and introduction are unclear and it is hard to infer the motivation and contribution. Since this is a novel problem, I encourage the authors to significantly improve the writing of the paper to convey the primary research contribution.

In intro:
>Labels are usually propagated conservatively: the output of an operation on data is labeled as the most restrictive (i.e., most confidential or least trusted) label of its inputs.

What work does this? There is no citation to this approach.

> Unfortunately, this approach of propagating the most restrictive input label to the output is too conservative for applications where LLMs operate on inputs retrieved from diverse sources.  The line in abstract does not read well. “Labels” are not defined at this point.


> The key idea behind our approach is to propagate only the labels of the samples that were influential in generating the model output and to eliminate the labels of unnecessary inputs.

“Samples” are not defined at this point

> improves the label in at least 50% of the cases on all datasets, thus mitigating the problem of label creep.

Unclear what improvement means here?

> We address this issue with the observation that it is sufficient to identify the labels that improve over the full context’s label, and consider only their corresponding subcontexts.

This line is not clear

 - The citations everywhere are not formatted correctly. Use \citep and \citet appropriately.

**Evaluation:**

- Why is only Llama-2 used for the evaluation? The authors should consider using more recent Llama-3 and more recent open-weight models.

- Dataset complexity: News article dataset only uses pairs of datasets which is High integrity and Low integrity. The QAs for the dataset were generated using GPT-4 and were not manually validated.  LLM agent dataset is manually curated but it also only has two labels. The proposed technique should be evaluated on a lattice that is more complex than just 2 classes.

- Figure-8 should also show a line for introspection baseline

---

> ### Author Response · Authors · 2025-08-07
> **Response to reviewer SC3C**
>
> We are very thankful to the reviewer for the useful and constructive feedback. We are glad to hear that the reviewer found the technical parts of the paper to be well-written and well-formalized. We address the points raised by the reviewer below:
>
>
> > Writing: abtract, citation format, and missing citation
>
> Thank you for highlighting these. We have fixed the citations as well as the abstract, based on the reviewer's suggestions.
>
>
> > Perform experiments with 3-4 more open-source models
>
> The algorithm discussed in the paper is not specific to the model architecture in any way. We describe a label propagation system that can propagate labels through any system, assuming it to be a black-box. Based on the reviewer’s suggestion, we have included additional results for 4 newer models (Mistral 7B, Llama-3.1 8B, Gemma-3 4B and Qwen-2.5 7B) on the LLM-agent dataset while using the same hyperparameters that we used before:
>
>
> | Model           | Label Prediction Method | Utility Target                  | Label Improvement | Missed Labels |
> |-----------------|------------------------|----------------------------------|-------------------|--------------|
> | Llama-2 (7B)    | Prompt-based           | Model generated $y$              | 85.0%             | 5.56%        |
> |                 | Introspection          | -                                | 45.0%             | 0.0%         |
> |                 | Prompt-based           | Ground truth $y^{\star}$         | 100.0%            | 0.0%         |
> | Llama-2 (70B)   | Prompt-based           | Model generated $y$              | 95.0%             | 9.53%        |
> |                 | Introspection          | -                                | 80.0%             | 11.1%        |
> |                 | Prompt-based           | Ground truth $y^{\star}$         | 100.0%            | 0.0%         |
> | Mistral (7B)    | Prompt-based           | Model generated $y$              | 97.6%             | 4.8%         |
> |                 | Introspection          | -                                | 100.0%            | 4.8%         |
> |                 | Prompt-based           | Ground truth $y^{\star}$         | 100.0%            | 0.0%         |
> | Llama-3.1 (8B)  | Prompt-based           | Model generated $y$              | 95.0%             | 17.4%        |
> |                 | Introspection          | -                                | 95.0%             | 17.4%        |
> |                 | Prompt-based           | Ground truth $y^{\star}$         | 100.0%            | 0.0%         |
> | Gemma-3 (4B)    | Prompt-based           | Model generated $y$              | 100.0%            | 9.1%         |
> |                 | Introspection          | -                                | 100.0%            | 7.0%         |
> |                 | Prompt-based           | Ground truth $y^{\star}$         | 90.0%             | 0.0%         |
> | Qwen-2.5 (7B)   | Prompt-based           | Model generated $y$              | 95.0%             | 9.5%         |
> |                 | Introspection          | -                                | 50.0%             | 9.1%         |
> |                 | Prompt-based           | Ground truth $y^{\star}$         | 100.0%            | 0.0%         |
>
>
> | Model           | ROUGE-L$(y, y^{\star})$ | ROUGE-L$(y', y^{\star})$ | Improvement possible | Not possible |
> |-----------------|------------------------|--------------------------|---------------------|--------------|
> | Llama-2 (7B)    | 0.77                   | 0.85                     | 0.0051              | -0.34        |
> | Llama-2 (70B)   | 0.82                   | 0.90                     | 0.025               | -0.31        |
> | Mistral (7B)    | 0.82                   | 0.90                     | 0.005               | -0.38        |
> | Llama-3.1 (8B)  | 0.76                   | 0.81                     | 0.001               | -0.22        |
> | Gemma-3 (4B)    | 0.83                   | 0.89                     | 0.03                | -0.39        |
> | Qwen-2.5 (7B)   | 0.81                   | 0.84                     | 0.002               | -0.30        |
>
>
>
> These results highlight that the proposed label propagation system is general and extends beyond any specific model architecture, but may require additional hyperparameter tuning in some cases.
>
>
> > Increase the complexity of the dataset and present individual error analysis for subset of the problems for both the baseline and the proposed technique.
>
> Our analysis focuses on different levels of complexity, starting with a very complex lattice with perfect ground-truth information (synthetic key-value dataset), moving to a more realistic natural language dataset with imperfect ground-truth (news article dataset), and finally shifting to the most timely application of our system, i.e., agentic workflows. While we acknowledge that these are synthetic setups, we believe that all these demonstrations capture the complexity and intricacy of any realistic application.

---

### Decision · Action_Editor_KXFd · 2025-09-18

**Recommendation:** Accept as is

**Audience:**

Yes

**Audience Explanation:**

The reviewers all found the results relevant to the TMLR audience.

**Claims And Evidence:**

Yes

**Claims Explanation:**

After discussion, all reviewers felt the claims are well supported.